



# A New Dual-Frequency Stratospheric Tropospheric / Meteor Radar: System Description and First Results

Qingchen Xu[1], Iain Murray Reid[2,3], Bing Cai[1], Christian Adami[2], Zengmao Zhang[1,4], Mingliang Zhao[1], Wen Li[1]

[1] State Key Laboratory of Space Weather, National Space Science Center, Chinese Academy of Sciences, Beijing, 100190, China
[2] ATRAD Pty. Ltd., 154 Ashley St., Underdale, 5032, Australia
[3] Department of Physics, School of Physical Sciences, The University of Adelaide, Adelaide, 5005, Australia
[4] College of Earth and Planetary Sciences, University of Chinese Academy of Sciences, Beijing, 100089, China

*Correspondence to*: Qingchen Xu (xqc@nssc.ac.cn)

**Abstract.** A new dual-frequency Stratospheric Tropospheric (ST) / Meteor radar has been built and installed at the Langfang Observatory in northern China. It utilizes a novel two-frequency system design that allows interleaved operation at 53.8 MHz for ST mode and at 35.0 MHz for Meteor mode, thus optimizing performance for both ST wind retrieval and meteor trail detection. In dedicated meteor mode, the daily meteor count rate reaches over 40,000 and allows wind estimation at finer time resolutions than the 1-hour typical of most meteor radars. The root mean square uncertainty of the ST wind measurements is better than 2 m/s when estimating the line of best fit with radiosonde winds. Preliminary observation results for one month of winter gravity wave (GW) momentum fluxes in the mesosphere, lower stratosphere and troposphere are also presented. A case of waves generated by the passage of a cold front is found.

## 1 Introduction

Mesosphere Stratosphere Troposphere (MST) radars operating in the VHF and UHF bands were originally hoped to provide a way of measuring winds, waves, and turbulence from heights close to the ground to heights close to 100 km (see e.g., Balsley and Gage, 1980; Chen et al., 2016; Qiao et al., 2020). The reality was that at the most common operating frequencies near 50 MHz, these radars, no matter how powerful, could only measure winds from near the ground to heights of around 20 km during the day and night (effectively an ST radar), and from 60 to 80 km during the day. The exception to this was for radars operating

in the polar regions in the summer, where strong radar returns were also received from a range of heights between 80 and 90 km (see e.g., Czechowsky et al., 1989; Morris et al., 2004).

Other techniques do offer a capability to measure dynamical parameters in the mesosphere and lower thermosphere (MLT) region. These include MF Partial Reflection (MFPR) radars (see Reid, 2015) and 'all-sky' meteor radars (see Hocking et al., 2001; Holdsworth et al., 2004). MFPR radars have largely, but not completely, fallen from favor for the reasons discussed by

Reid (2015), while meteor radars have become very widely applied (see e.g., Koushik et al., 2020; Luo et al., 2021).





By combining a meteor capability within an ST radar operating at a single frequency, a radar with MST measurement capabilities can be created. This was initially applied to narrow beam ST radars (see e.g., Cervera and Reid, 1995; Valentic et al., 1996), and subsequently by adding an 'all-sky' capability to an existing ST radar (see e.g., Reid et al., 2006). The height resolution in meteor mode in the mesosphere is not as good as for a true MST radar (~2 km versus ~300 m), but dynamical
information is typically available both day and night in the 75 to 110 km height region.

This opens the possibility of studying the dynamics of both the troposphere stratosphere (ST) and the mesosphere lower thermosphere (MLT) regions with the one radar. Understanding the dynamical coupling between the lower and upper parts of the atmosphere is an essential part of atmospheric dynamics. A significant contributor to coupling from below is due to the upward propagation of internal atmospheric gravity waves generated in the lower atmosphere which break in the upper
atmosphere thereby depositing momentum and energy. These wave motions can be characterized in a statistical sense by calculating the (density normalized) Reynolds Stress Tensor. This contains the essential dynamical information related to both density normalized kinetic energy, viz., $(\overline{u'^2}, \overline{v'^2}, \overline{w'^2})$, which represent the zonal, meridional and upward kinetic energies, and density normalized momentum transport, viz., $(\overline{u'w'}, \overline{v'w'}, \overline{u'v'})$, which represent the upward transport of zonal and meridional momentum, and the horizontal transport of momentum respectively. These terms are available in the lower
atmosphere (except for $\overline{u'v'}$), by directly by using the five beams of the ST radar, and in the upper atmosphere from the numerous radial velocities and angle of arrivals (effectively 'radar beams') of the meteor radar. The variation of these terms with height reveals the transfer of energy and momentum from the wave motions into the background wind. For example, the divergence of the $\overline{u'w'}$ term with height is a measure of the zonal mean flow acceleration. We discuss this in more detail below.

For ST operation, a frequency near 50 MHz is typical, while for meteor operation a frequency near 30 MHz is typical. Since
the meteor radar is the 'add-on', these single frequency ST / Meteor radars operate at the ST radar frequency. This is important in terms of performance as is evident if we consider the expression for count rate given by McKinley (1960). This gives the meteor count rate $N$ dependence on transmitted power $P_T$, radar wavelength $\lambda$ (essentially the reciprocal of the operating frequency), system gain $G$ and received power $P_R$ as follows:

$$N \propto \frac{P_T^{1/2} G \lambda^{3/2}}{P_R^{1/2}} \tag{1}$$

Inspection of equation (1) indicates that meteor counts are proportional to $\lambda^{3/2}$, so operating a meteor radar near 50 MHz
results in fewer counts than for an equivalent 30 MHz radar. This is typically compensated by the higher powers available on the ST radar, noting that counts are proportional to $P_T^{1/2}$. Naturally, interleaving the operating modes does reduce overall meteor count rate.

There have been several single frequency combination ST / Meteor radars in operation. These include the Wakkanai ST radar (Ogawa et al., 2011; although this radar did not exercise the meteor option), the Davis ST radar (Reid et al., 2006), the Kunming
ST radar (Yi et al., 2018), and the Buckland Park ST (BPST) radar (Reid et al., 2018). However, only the BPST radar is





currently run routinely in interleaved meteor / ST mode. An obvious extension of this single frequency approach is to operate the radar at two frequencies optimized for each operational mode. In this paper, we describe such a radar and its first results.

## 2 Radar System

### 2.1 Overview

In 2010, the National Space Science Center of the Chinese Academy of Sciences (NSSC) installed an 'all-sky' interferometric meteor radar at the Langfang Observatory (39.39º N, 116.66º E), Hebei Province (e.g., Tian et al., 2021). This radar operated at a frequency of 35.0 MHz with a peak power of 20 kW, using a vertically pointing 2-element Yagi antenna and five individual 2-element receiving antennas arranged as a cross shaped interferometer, with horizontal distances of 2 or 2.5 wavelengths between antennas (the so-called "JWH configuration" (e.g., Jones et al., 1998)), respectively. In cooperation with ATRAD,

NSSC replaced this radar with a new combination ST / Meteor radar during the period from 2018 to 2021. A unique feature of this new radar system is that two frequencies are used in interleaved operation: 53.8 MHz for ST mode and 35.0 MHz for Meteor mode, thus optimizing performance for both ST wind retrieval and meteor trail detection. Table 1 summarizes the basic radar parameters.

**Table 1: Basic parameters of new dual-frequency ST / meteor radar**

| Parameter | Value |
|---|---|
| Location | NSSC Langfang Observatory (39.39° N, 116.66° E), China |
| Operating frequency | 35.0 MHz in Meteor Mode and 53.8 MHz in ST Mode |
| Peak power | 48kW |
| Maximum duty cycle | 10% |
| Transmitted waveforms | Single pulse, complementary, Barker and user defined codes |
| Pulse shapes | Gaussian; Raised cosine |
| Antenna | Meteor Antennas: One 2-element Yagi for Tx, and Five 2-element Yagis for Rx, tuned for 35.0 MHz ST Antennas: 144 (12x12) 3-element Yagis for Tx/Rx, tuned for 53.8 MHz |
| Receiving channels | 6 (5 meteor receiving channels, 1 ST receiving channel) |
| Observing mode | ST Low Mode, ST High Mode, Meteor Mode, Interleaved Mode |
| ST mode beams | 15° off-zenith towards the N, S, E and W, and V |

Installation work on the new radar started in November 2018 with field site electromagnetic environment measurements. The manufacture and factory testing of all the new system modules were finished in January 2020, and the old meteor radar was switched off in April 2020. Because of COVID-19, infrastructure construction, radar module installation and system





integration were delayed, with the initial radar site test completed in September 2021. Final system signoff occurred after more than three months of test operation in December 2021.

## 2.2 System Description

The two basic radar types included in this combined system have been installed at numerous locations worldwide. This includes more than 25 of the meteor radars, and more than 20 of the ST radars. The basic ST radar and hardware is described in Dolman

et al. (2018). The meteor radar approach is described by Holdsworth et al. (2004), noting that the power amplifiers and transceiver are rather more advanced than those described by these latter authors. The new feature of this execution is that the radar uses twelve 4 kW power amplifiers that are common to both ST and meteor operation, and which operate at both 53.8 MHz and 35.0 MHz.

The photos of the new dual-frequency ST / meteor radar are shown in Figure 2. Outdoor Antennas include ST antenna array,

meteor transmitting (Tx) antenna and receiving (Rx) array. DBS rack, Tx rack and Rx rack are laid from left to right in the radar hut.

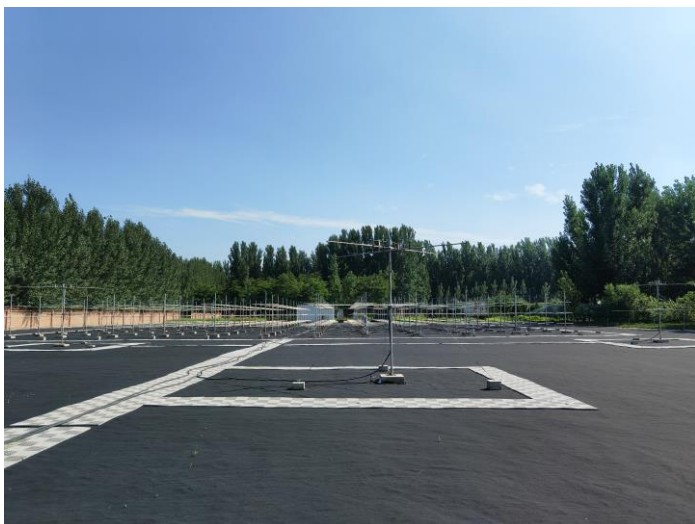
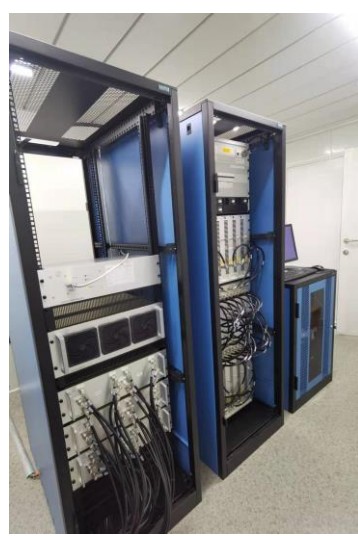

**(a) Outdoor antennas**                                          **(b) DBS rack (left), Tx rack (middle) and Rx rack (right)**

**Figure 1: The Photos of the new dual-frequency ST / meteor radar**





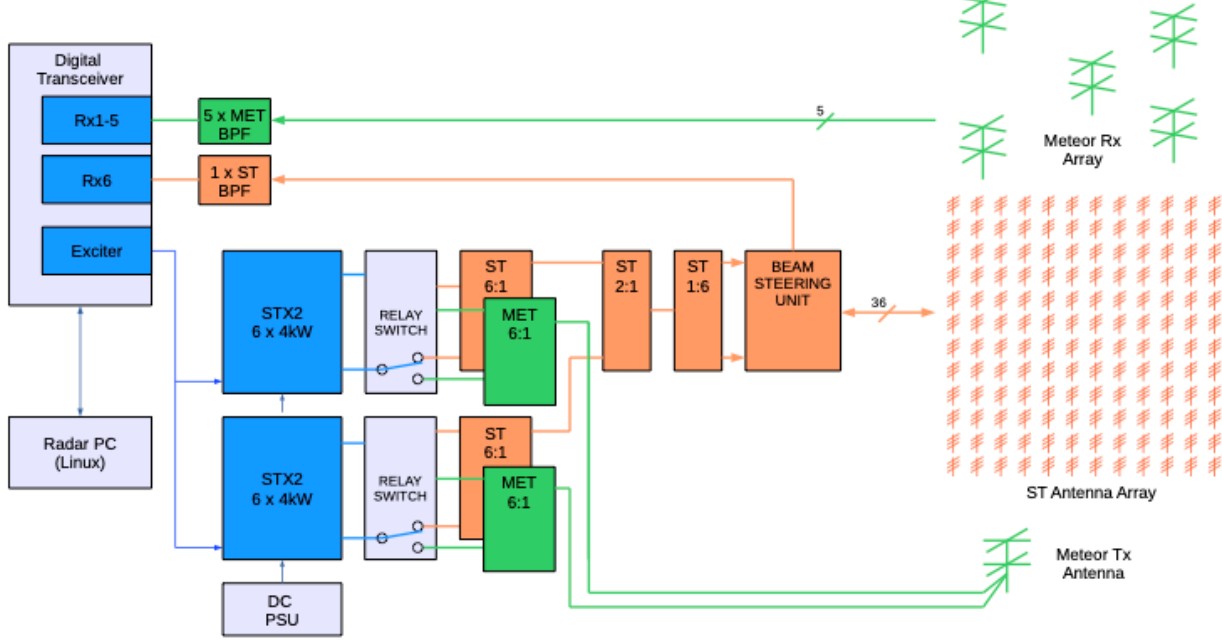

Note:
1. Dual frequency 35 and 53.8 MHz
2. Meteor 35 MHz
3. ST 53.8 MHz


**Figure 2: The block diagram of the new dual-frequency ST / meteor radar**

The block diagram of the new dual-frequency ST / meteor radar is shown in **Figure 2**. The sophisticated dual-frequency transmitter includes two sets of STX2 solid Transmitters each composed of six 4 kW Power Amplifier (PA) modules, two sets

of Single Pole Double Throw (SPDT) High Power (HP) Relay Switches, one 53.8 MHz High Power (HP) 12:2 Combiner, one 53.8MHz HP 2:1 Combiner and one 35.0 MHz HP 12:2 Combiner. Each PA can be operated both in 35.0 MHz and 53.8 MHz, and is connected to a SPDT HP Switch. The modulated radio frequency (RF) pulse and trigger signal for the transmitter are generated by the exciter of the digital transceiver, then amplified by twelve 4 kW PA modules. Then twelve 4 kW pulses are switched to 53.8 MHz HP 12:2 Combiner and then combined into a single 48 kW peak power RF output for Doppler Beam

Steering (DBS) operation in ST mode, or switched to 35.0 MHz HP 12:2 Combiner and combined into two 24 kW peak power RF outputs in Meteor mode.

The ST antenna array consists of a square grid formed by 144 3-element Yagi antennas, and has the 3 dB full-width of 6.7 degrees. On transmission in DBS mode, switching and appropriate phase delays are used to generate a vertical beam and four off-zenith beams (north, east, south, and west) with a tilt angle from the zenith of 15 degrees. On reception, signals returned

from the atmosphere are combined into one signal and fed into the digital transceiver. In meteor mode, the "JWH configuration"





design is utilized with the entire power transmitted through a single crossed folded dipole, with half the power delivered to each arm.

A flexible 6-channel digital transceiver (an exciter / receiver) is configured so that five channels are dedicated to meteor mode and one channel to ST mode. Echoes are first filtered and amplified, mixed to an Intermediate Frequency (IF) band, and

digitized and down converted to baseband for further analysis.

The radar PC manages the radar and monitors system status in real time. Three operating modes are typically used: ST Low Mode, ST High Mode, and Meteor Mode. These are interleaved in the radar's regular configuration. In ST mode, the sampling range is usually from 300 m to 20 km. The pulse width and range sampling resolution are adjustable from 100 m to 4000 m; the pulse repetition frequency (PRF) is up to 200 kHz; and 3 or 5 beams can be set to achieve different time resolutions of

wind profiles from 3-min to 1-hour.

In meteor mode, the PRF is much lower at 430 Hz, allowing an unambiguous sampling range up to 300 km. The pulse width is set to no less than 1.8 km and coded pulses are usually adopted. While the range sampling resolution remains 1.8 km, the time resolution of meteor wind profiles can be set to 15-min, 30-min and 1-hour depending on the meteor count rate because of the very high-count rate. The validity of these shorter averaging periods will be investigated in future work.  The new radar

has been operated in interleaved ST Low Mode, ST High Mode and Meteor Mode Since November 9th, 2021, and the main operating parameters are shown in **Table 2**.

**Table 2: Operating parameters for radar observations**

| Parameter | ST Low Mode | ST High Mode | Meteor Mode |
|---|---|---|---|
| **PRF (Hz)** | 14000 | 6000 | 430 |
| **Transmit pulse HPFW (m)** | 100 | 600 | 7200 |
| **Pulse code type** | None | None for these observations | 4-bit complementary |
| **Pulse shape** | Gaussian | Gaussian | Gaussian |
| **Range (km)** | 0.3 ~ 8 | 1.2 ~ 22.2 | 68.4 ~ 318.6 |
| **Range sampling resolution (m)** | 100 | 600 | 1800 |
| **Coherent integrations** | 700 | 150 | 4 |
| **Beams** | 5 | 5 | - |

## 3 Radar Performance

The first stage of the installation and system integration work was completed in March 2021 when a malfunction of the 53.8 MHz HP Combiner was found. The damaged combiner was repaired and reinstalled in September 2021, followed by an intensive ST wind measurement test and initial data validation. A thorough system test was conducted from November to the end of December 2021, and data validation work was completed in February 2022. First results using the observational data

accumulated from March 2021 to January 2022 are presented here to demonstrate the performance and functionality of the new radar.





### 3.1 Meteor Radar

A good opportunity to investigate the meteor detection capability of this new radar was presented when the 53.8 MHz HP
Combiner was under repair. The radar was run intermittently in dedicated meteor mode (with the parameters shown in **Table**
**2**) from March to the beginning of September as the infrastructure construction continued. There were three observational
periods (OPs) of dedicated meteor mode. These were 03/16/2021 to 04/14/2021; 06/16/2021 to 06/30/2021; and 07/20/2021
to 09/07/2021. The first two were relatively continuous observation periods; OP3 involved several system halts, for example,
from 07/29/2021 to 08/04/2021.

The daily meteor count rate reached over 40,000 in most complete observing days (see Figure 3), and surpassed 60,000 (Figure
3 and Figure 4) at the beginning of September. Such high counts allow wind estimation of finer time resolutions (e.g., 30-min)
rather than the 1-hour typical of most meteor radars. Winds were calculated as described by Holdsworth et al. (2004).
Representative examples of horizontal winds calculated with different time intervals are shown in Figure 5 and Figure 6.
Inspection of these figures indicates that wind measurements between ~80 km to 100 km exhibit good continuity with 30-min
intervals and show more detailed variations than the 1-hour interval data, opening the possibility of investigating shorter period
motions such as gravity waves and turbulence.

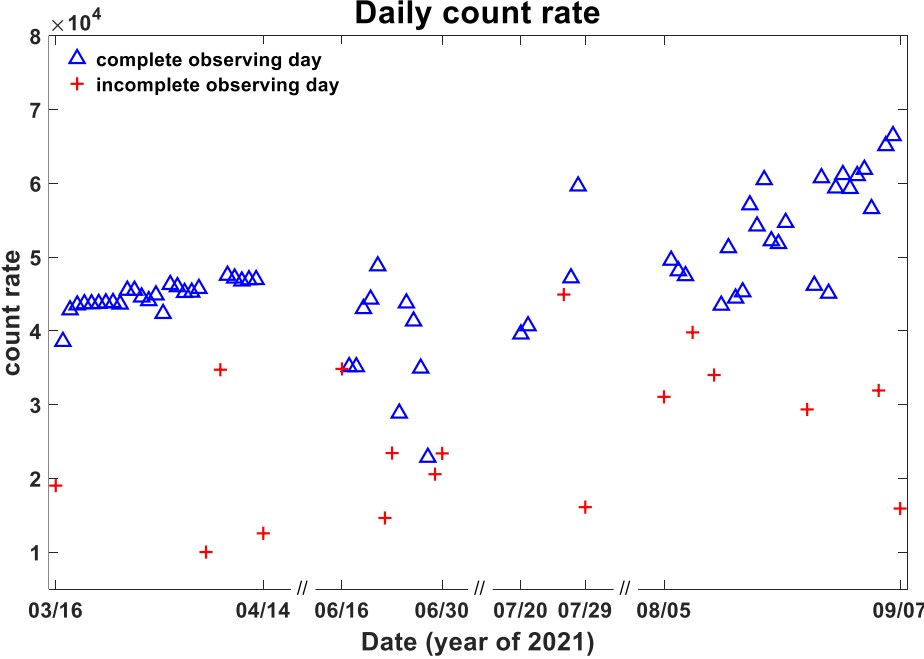

**Figure 3: Daily count rate during three observation periods when radar was operated in dedicated meteor mode. Count rates under
10,000 due to radar halts are not shown.**






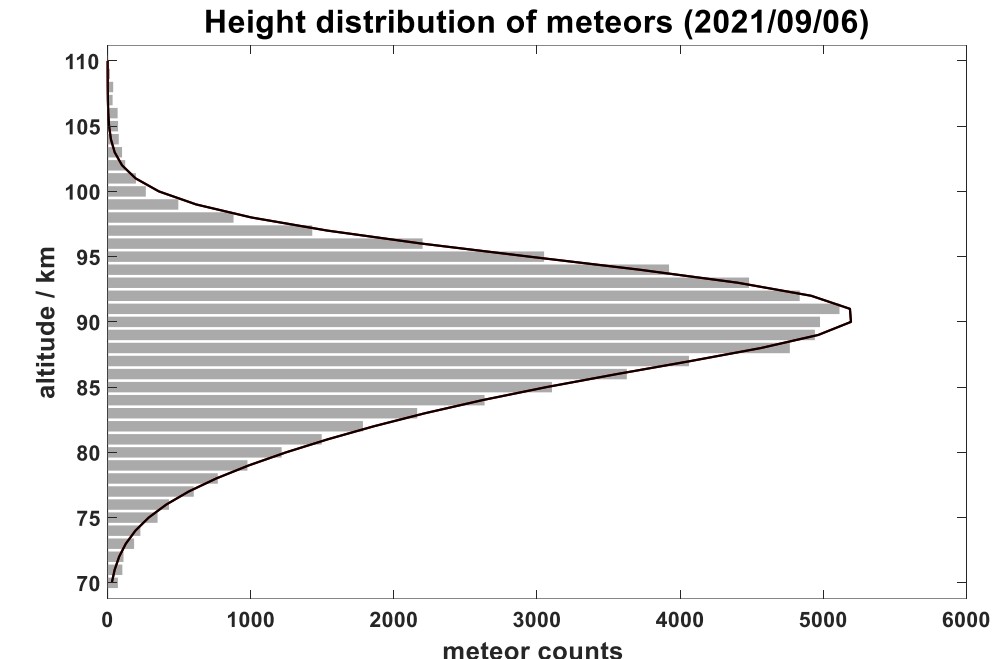

**(a) Height distribution**

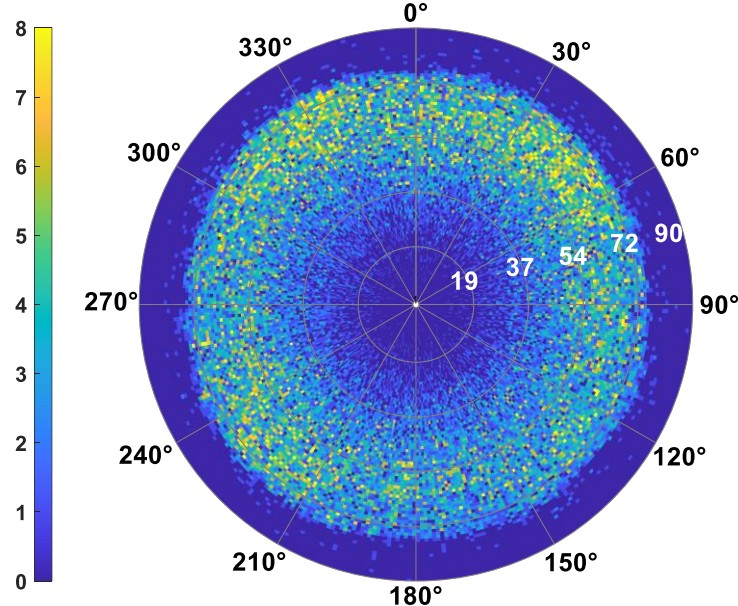

**(b) Azimuthal and zenithal distribution**

**Figure 4: Meteor echoes on 09/06/2021, including 66458 underdense meteors: (a) Height distribution using 1 km vertical gates and (b) Azimuthal and zenithal distribution with grid of 1 degree.**





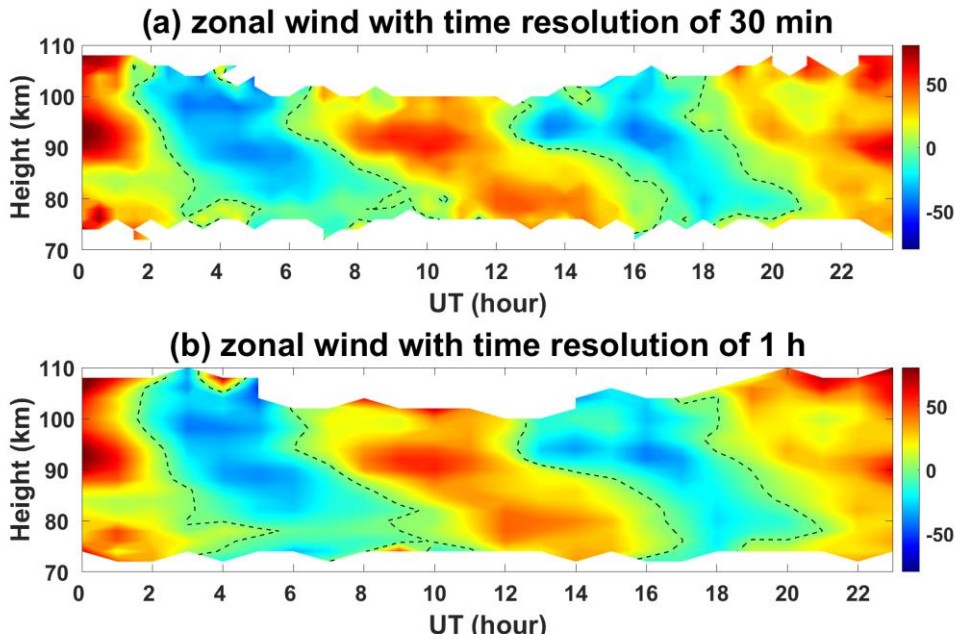

Figure 5: Zonal wind on 09/06/2021 with (a) 30-min interval and (b) 1-hour interval

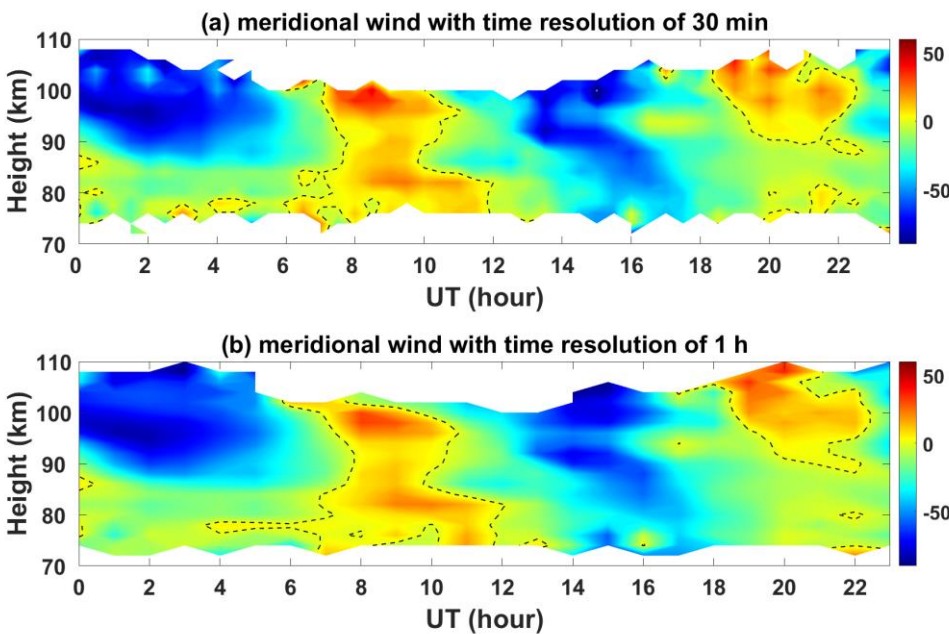


Figure 6: Meridional wind on 09/06/2021 with (a) 30-min interval and (b) 1-hour interval



## 3.2 Meteor Wind Comparisons

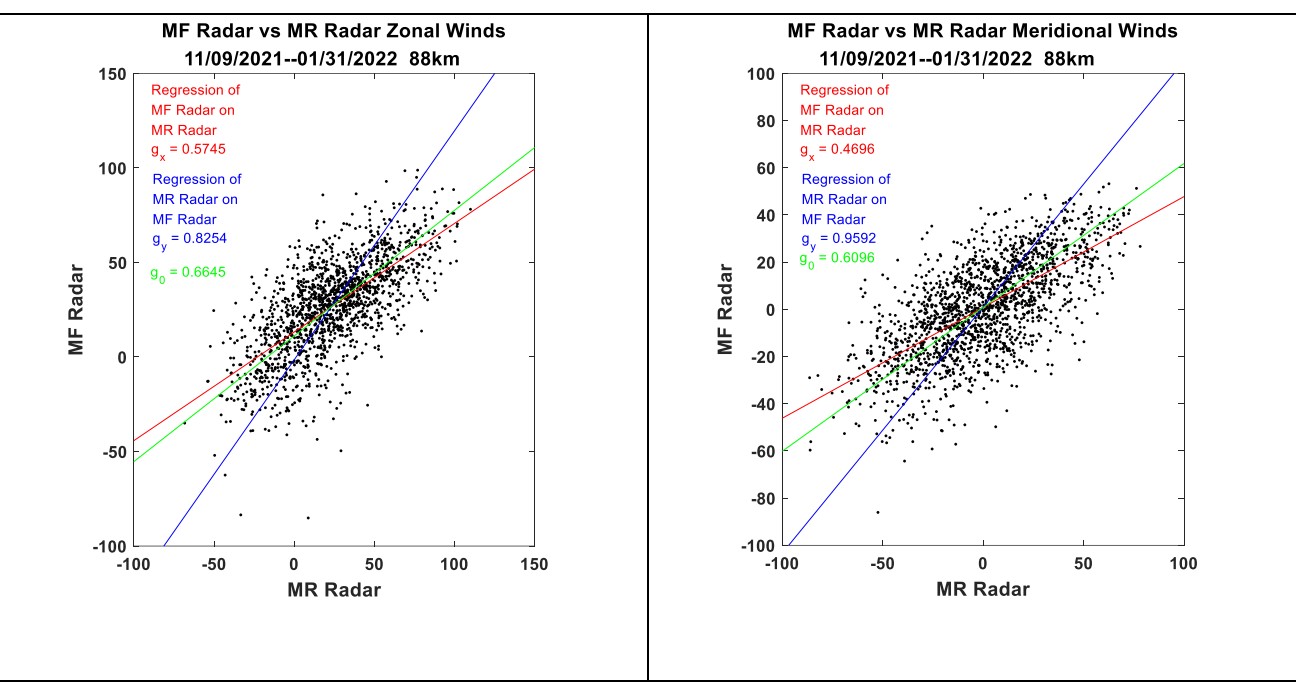

**Figure 7: Intercomparison of MFPR and Meteor radar zonal (left) and meridional winds (right) for a height of 88 km.**






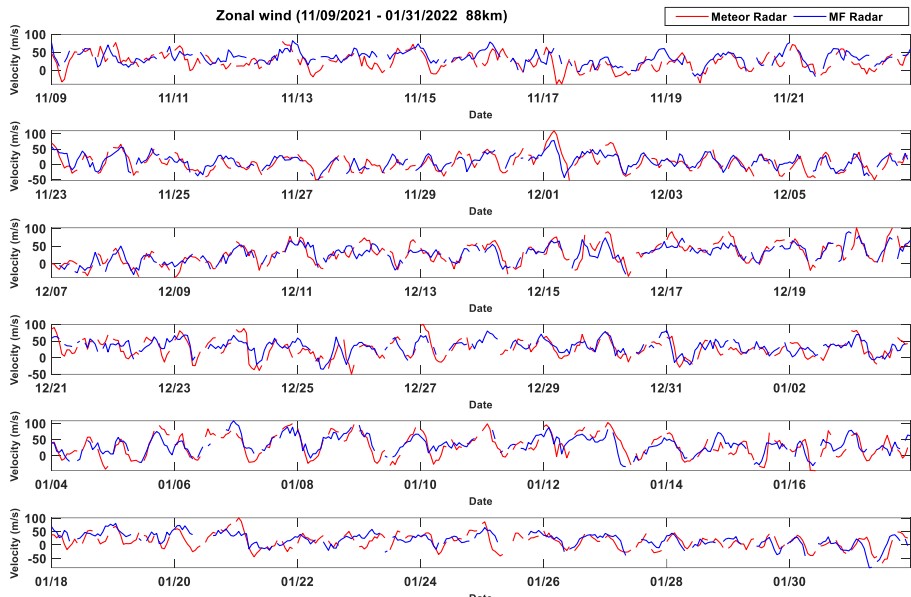

**Figure 8: Time series of zonal wind from MFPR and Meteor radar for a height of 88 km**

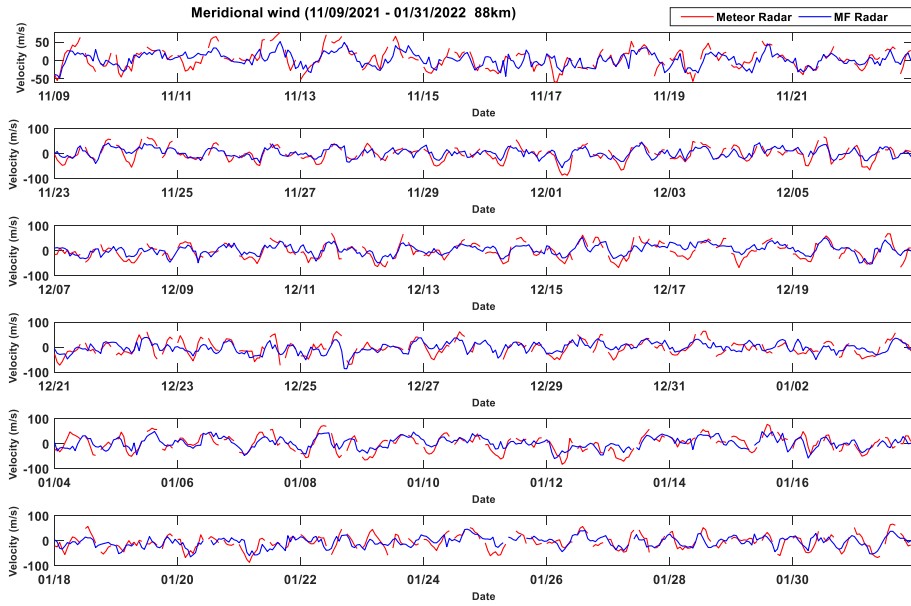

**Figure 9: Time series of meridional wind from MFPR and Meteor radar for a height of 88 km**





An MFPR radar has operated at the Langfang Observatory since 2009 (see e.g., Cai et al., 2021). Notwithstanding the known bias towards wind underestimation in uncorrected MFPR winds, which is found to be up to 15 to 40 % and strongly height dependent and probably caused by any noise in the full correlation analysis (FCA), so-called triangle size effect (TSE), low

sample rate and so on (see e.g., Reid, 2015), it is interesting to do a quick intercomparison here in anticipation of a longer future investigation. Figure 7 shows the MFPR and meteor radar winds for a height of 88 km for the one-month period of observation. Because there are errors in each technique, the 'gain factor' $g_0$ is calculated following Hocking et al. (2001) to account this. In the case of the zonal wind component, this results in a slope of 0.67, so that the MFPR winds are smaller than the meteor winds by about 1.49. This is similar to results found by numerous authors (see e.g., Reid et al., 2018) and can be

used to correct the MFPR winds as the phases are consistent between the two techniques (see Figure 8 and Figure 9).

**3.3 Stratosphere-Troposphere (ST) Radar**

On November 9th, 2021, the new radar was configured for system testing, and was run as follows. Interleaved 5-beam ST low/high mode runs from 16:30-18:10, 10:30-12:10 and 22:30-00:10 UT (matching a radiosonde launch schedule). Outside of these intervals, meteor mode ran for 20 minutes (10 minutes to 30 minutes past the hour), and interleaved ST low/high mode

ran for 40 minutes (30 minutes past the hour to 10 minutes past the next hour). Winds were calculated using radial velocities from 5 beams. The off-zenith angle of 15° is chosen to minimize the effects of aspect sensitivity on the relatively broad beams. This approach has been validated on numerous radars, including the Australian Wind Profiler Network, which incorporates four ST wind profilers of basically the same design as the present ST section of the Langfang radar (Dolman et al., 2018). This network was also used to validate Aeolus satellite results over Australia (Zuo et al., 2022).

The height distributions of successful ST wind measurements for both high and low mode are shown in Figure 10. For this observational period, 498 is the maximum profile number that could be obtained during system testing for both high and low mode. In high mode, observations begin at 1.2 km and extend to heights up to near 22 km, although there are very few of the latter. In low mode, observations begin at 300 m and extend up to heights near 8 km. For the remainder of the current work, we will focus on the high mode observations, and for those heights that have acceptance rates over 50 %.






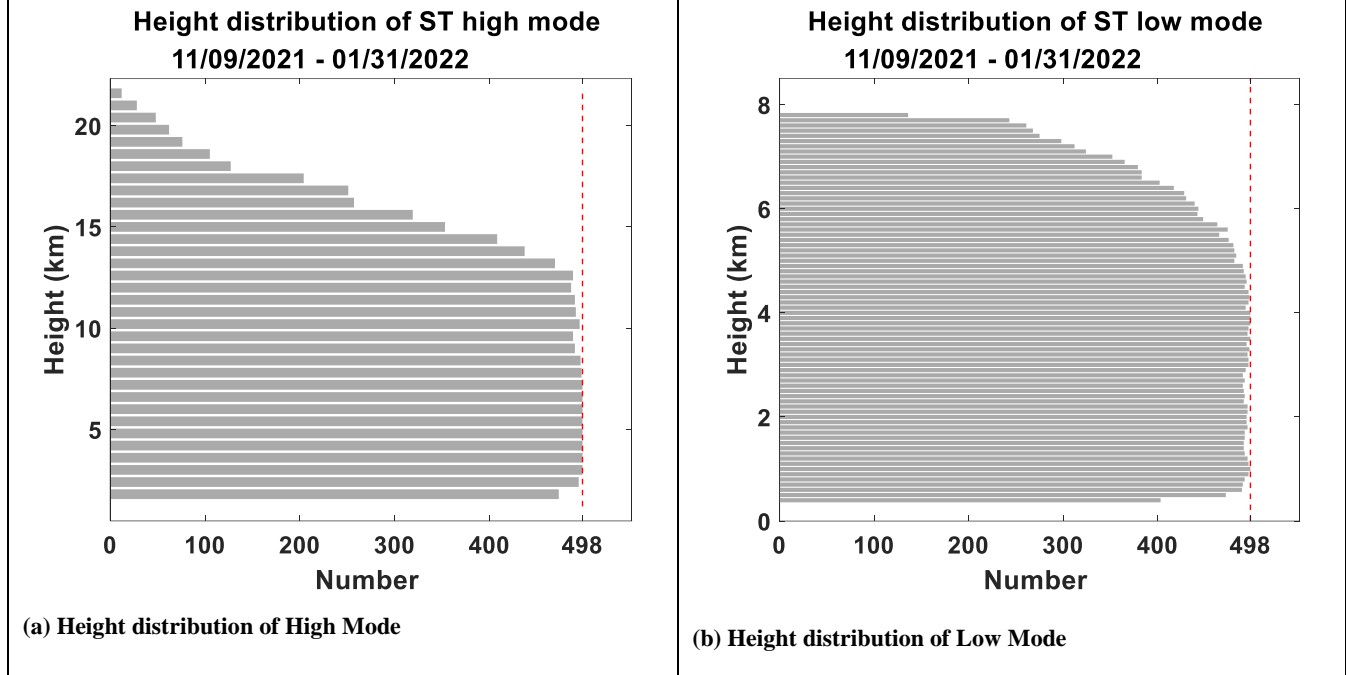

**Figure 10: Height distribution of ST wind measurement for (left) high mode and (right) low mode**

Acceptance rates for High Mode for the three wind components are shown in Figure 11(a). These are about 50% near 16 km. The mean tropopause height determined from nearby radiosonde observations is near 9.6 km and is shown in this figure as a dashed pink line. The mean winds for the entire observational period are shown in Figure 11(b), along with the standard deviation of the wind over that period.





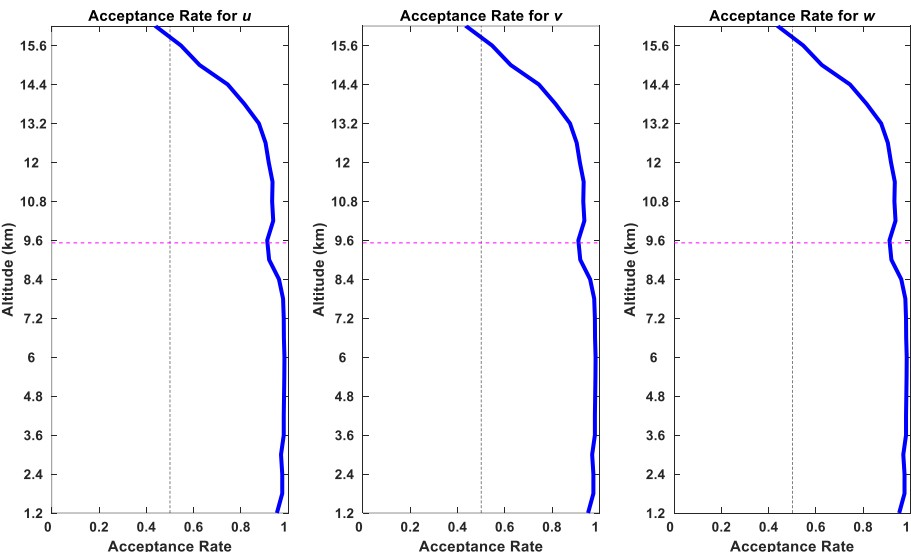

**(a) The acceptance rates for the background wind estimation, which are the ratios of the number of successfully retrieved**
**winds to the number of acquired raw data.**

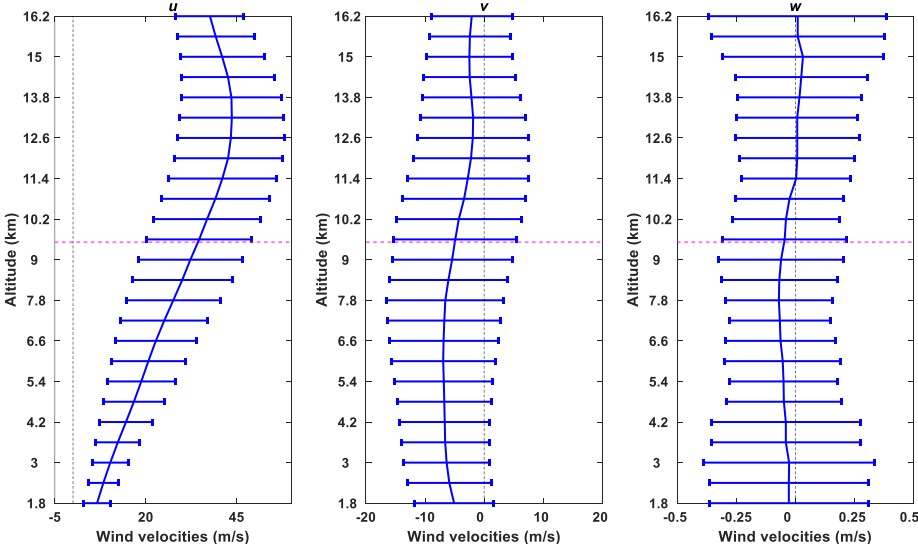

**(b) Monthly-averaged height variations of mean zonal, meridional, and vertical winds along with respective standard deviation profiles, pink line indicated tropopause height.**

**Figure 11: Acceptance rates and the monthly background winds. The pink line indicates the radiosonde tropopause height.**






### 3.3.1 ST Wind Comparisons

Nearly three month's radar observation profiles of 30-min time interval (11/09/2021 to 01/31/2022) and radiosonde measurements were used to evaluate the reliability and accuracy of the ST wind measurements. Radiosondes are regularly launched at 11:15, 17:15 and 23:15 UT each day from the Beijing Meteorological Observatory (39.80° N, 116.47° E), station

index number 54511, which is about 50 km north of the Langfang Observatory. Radiosonde data are acquired from the GTS1 type digital radiosonde and the horizontal winds are obtained by tracking the position of the balloon using L-band radar (e.g., Li et al., 2011). The raw data are sampled with 1-s interval, resulting in an uneven height resolution.

Radar profiles within the period of from 30 minutes before the sonde launch and 30 minutes after the sonde launch were selected. One sonde profile matched two (occasionally one) radar profiles and made two profile pairs. Sonde data were

averaged spatially to match the radar range sampling interval in both low and high modes. Radar profiles were quality-controlled using the five-point center moving average method (see Tian et al., 2017) to remove outliers mainly produced by air traffic. In addition, data corresponding to returns with signal-to-noise ratios below -12 dB were rejected. No further attempts were made to remove outliers in the radar profiles or errors in the sonde data. All available data pairs were then used to estimate the line of best fit (see Dolman et al., 2018). Comparisons for the high mode zonal and meridional winds are shown in Figure

12. The slopes of best fit lines attributing root mean square (rms) errors of 0.15, 0.5, 1.0 and 2.0 to the sonde are summarized in Table 3, along with the number of profile pairs and data points.

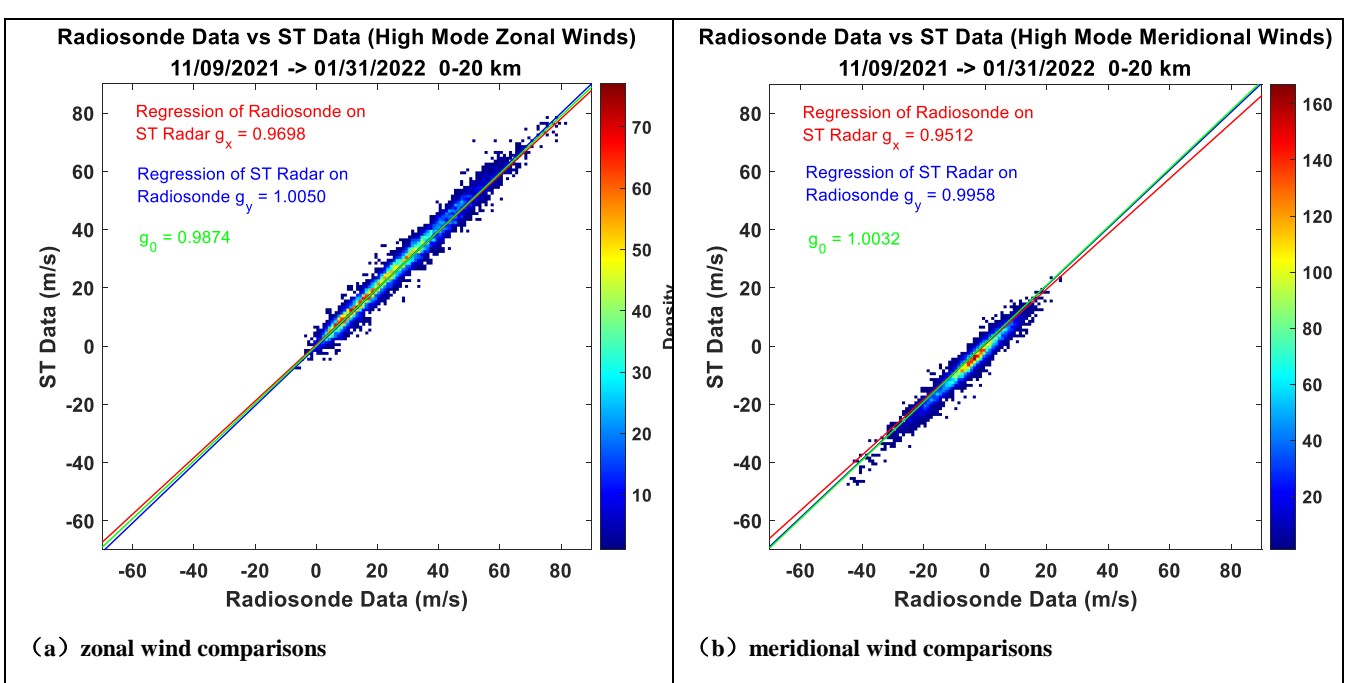

**Figure 12: Zonal (a) and meridional (b) colored contour wind comparisons for ST High Mode. The sonde data are shown on the x axis, with radar data on the y axis, and colors indicate data density.**




Table 3 demonstrates that both low and high modes are in good agreement as the slopes of best fit lines lie in 0.98 ~ 1.01 when attributing rms errors of 2.0 $ms^{-1}$ to the sonde, which might also suggest the actual uncertainty of sonde data.

**Table 3: The slopes of best fit lines for zonal and meridional wind comparisons**

| Observation Mode | Comparing profile pairs | Comparing data points | | Attributing errors to the sonde | | | |
|---|---|---|---|---|---|---|---|
| | | | | 0.15 | 0.5 | 1.0 | 2.0 |
| **ST High** | 498 | 11761 | Zonal | 0.970 | 0.971 | 0.974 | 0.987 |
| | | 11797 | Meridional | 0.952 | 0.954 | 0.964 | 1.003 |
| **ST Low** | 498 | 31918 | Zonal | 0.959 | 0.962 | 0.970 | 1.006 |
| | | 32212 | Meridional | 0.922 | 0.925 | 0.936 | 0.982 |

### 3.3.2 Radiosonde Temperatures

In addition to the wind velocity, the twelve hourly interval radiosonde data have been used to obtain temperature and density profiles. We note that the midnight flights only include wind data up to less than 20 km. We have estimated the tropopause

height using the radiosonde data. Temperatures for the one-month observational period and the tropopause height are shown in Figure 13.

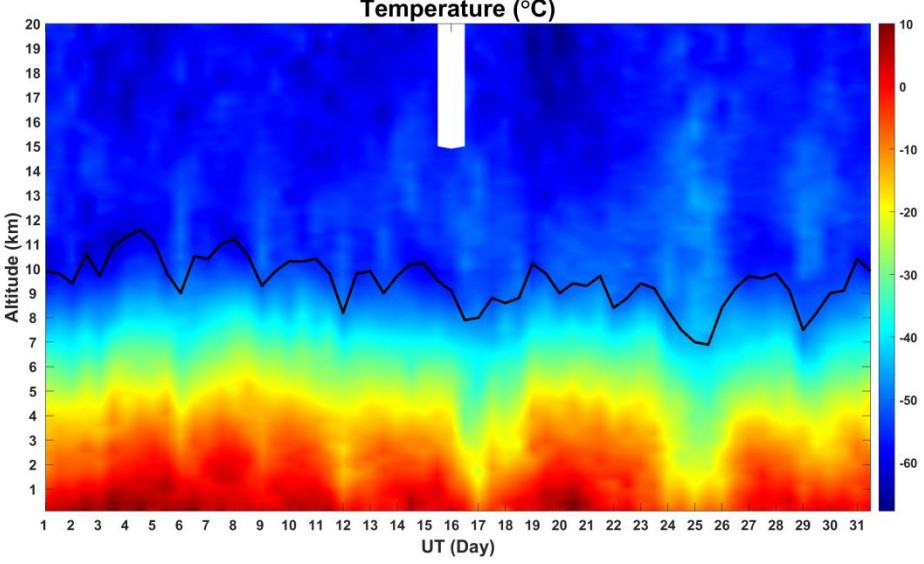

**Figure 13: Temperature profiles from radiosonde data, black line indicated tropopause height.**




# 4 Preliminary Estimation of Gravity Wave Momentum Flux

Gravity waves (GWs) play an important role in middle atmospheric dynamics and energetics. Most GWs are generated in the troposphere, and transfer and deposit their momentum in the middle atmosphere when propagating upward. The new radar, with good height coverage and time resolution in both the lower and upper atmosphere, giving a true MST capability, permits a simultaneous investigation of the momentum transport in both regions. Here we present preliminary results of gravity wave momentum flux in the troposphere, lower stratosphere, and mesosphere utilizing one month's radar observation data (12/01/2021 to 12/31/2021). We begin with the Stratospheric Tropospheric observations.

## 4.1 Stratospheric Tropospheric Winds

The 30-min averaged high-mode wind components for the month of observations are shown as height-time plots in Figure 14. The tropopause height has been overplotted. Zonal wind speeds are generally positive, exhibit a clear jet-like structure and reach values close to 80 m/s near the peak of the jet. Meridional wind speeds vary between ± 50 m/s. There are clear shorter-term variations superimposed on the longer-term wind speed variations in both the zonal and meridional wind components. Lower tropopause temperatures are generally associated with southward winds. Vertical velocities lie in the ±1.5 m/s range and also show considerable shorter period variations. These are clearer in all three wind components when periods longer than the inertial period of 18.9-h are filtered out as shown on the right-hand side of this figure. Inspection of these plots shows wave activity across the entire observational period and across all heights. Phase fronts tend to be vertical, indicating non-propagating waves, but some wave fronts are tilted, indicating upward or downward propagation. There are several periods of intense wave activity evident in both horizontal wind components. This is particularly so on Dec. 3rd, Dec. 5th to 6th, Dec. 8th to 9th, especially in the filtered meridional winds and during periods of southward winds. Very strong waves are evident in both horizontal wind components with the passage of a cold front on the 16th of Dec. These extend across the tropopause but are strongest below it. This period also corresponds to the highest values of the zonal jet speed. The cold front is associated with strong southward and downward vertical winds and a temperature decline (see Figure 13 and Figure 14). This associated oscillation has a period of around half a day and is apparently attenuated away from the tropopause, but evident over the entire height region of the observations. It continues through the Dec. 17th to 20th.

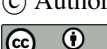

**(a) 'Raw' zonal wind**

**(b) Filtered zonal wind**

**(c) 'Raw' meridional wind**

**(d) Filtered meridional wind**

**(e) 'Raw' vertical wind**

**(f) Filtered vertical wind**

**Figure 14: 'Raw' and 'filtered' (for periods less than 18.9 h) background winds from 30-min averaged ST High Mode observation data, black line indicated tropopause height.**





### 4.2 Gravity Wave Momentum Flux and Vertical Transport in the Troposphere Lower Stratosphere Region

The dual complementary coplanar beam method (see e.g., Vincent and Reid, 1983) is adopted to make direct measurements of gravity wave momentum flux in the troposphere and lower stratosphere (TLS) region. The difference between the mean square fluctuating radial velocities of two symmetry beams pointing at zenith angles $+\theta$ and $-\theta$ is obtained to calculate the vertical flux of horizontal momentum. For zonal components of the momentum flux $\overline{u'w'}$,

$$\overline{u'w'} = \frac{\overline{v_E^2} - \overline{v_W^2}}{2sin2\theta_E} \tag{2}$$

A similar expression applies for the meridional component of the momentum flux.

Here $v_E$, $v_W$, $v_N$, $v_S$ represent the radial perturbation velocities in the east, west, north, south beams, and $u'$, $v'$, $w'$ refer to the fluctuating zonal, meridional, vertical winds, respectively. $\theta_E$ in equation (2) is the effective beam direction for the off-zenith beams, which should replace the apparent off-vertical angle $\theta_A$ considering the influence of aspect sensitivity (Reid et al, 2018b). As we have seen above, the mean winds evaluated assuming that the apparent and effective beam angles are the same are in excellent agreement with the radiosonde measurements in the present study, and with numerous other intercomparisons

made with this ST radar type. However, equations (2) and (3) involve differencing two like quantities to obtain a small quantity and so we apply the beam direction corrections to these measurements to ensure valid values.

### 4.2.1 Momentum Flux

A 5-beam time series of radial velocities with an equivalent sampling interval of 10 minutes have been used to retrieve the upward horizontal fluxes. Outliers such as aircraft echoes and ground clutter were removed by taking the mean velocity in a

3-hour sliding window for each height interval and discarding values exceeding three standard deviations from the mean. A spline was applied to fill in missing data points. The time series of perturbation profiles then were filtered using a 5[th]-order Butterworth high pass filter to retain oscillations with period less than the inertial period at this latitude (18.9-h). Variances were finally calculated for the filtered time series, using only values corresponding to those times when real data were obtained. Estimates of the horizontal momentum fluxes are shown in Figure 15 and Figure 16.

From Figure 15 we can see the cold front beginning on Dec. 16[th] is associated with large values of the density normalized fluxes. We note the strong waves evident in both the eastward and northward winds in the troposphere at this time.

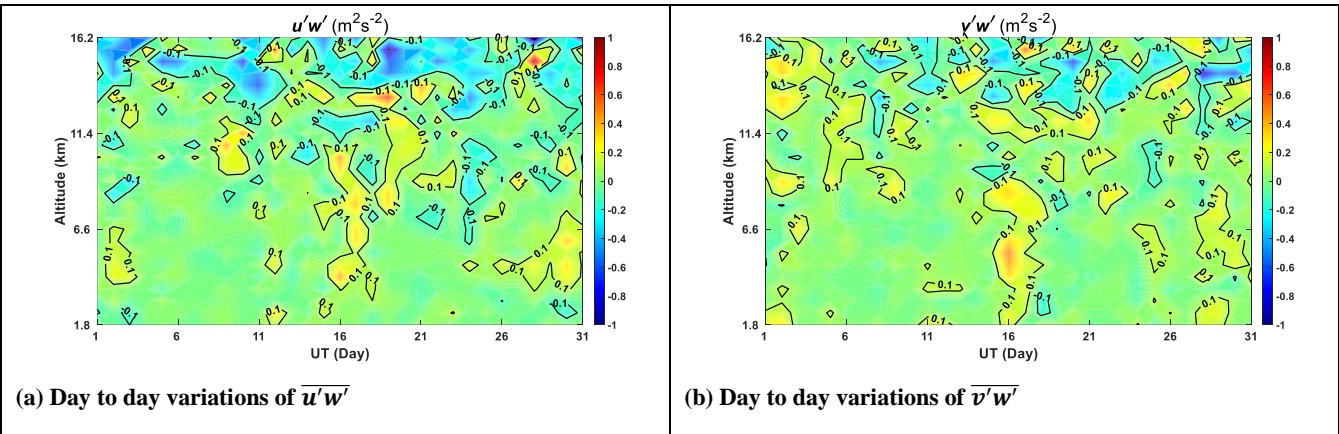

**(a) Day to day variations of $\overline{u'w'}$**          **(b) Day to day variations of $\overline{v'w'}$**

**Figure 15: Density normalized momentum fluxes**

Also from Figure 15 we can see that horizontal momentum fluxes have clear day-to-day variations especially during the surface weather process. The zonal and meridional components reach values near 0.4 Pa at 4.2 km and 5.4 km respectively on Dec. 16th associated with a cold front (see Figure 13), and mostly range between ± 0.05 Pa in calmer weather. The mean density normalized fluxes are shown in Figure 16. The monthly-averaged meridional momentum fluxes are predominately northward, opposite to the mean meridional winds which are dominated by the weak southward flow. The monthly-averaged zonal

momentum fluxes are near to zero in the troposphere and predominately westward when entering the stratosphere, opposite to the mean eastward flow.

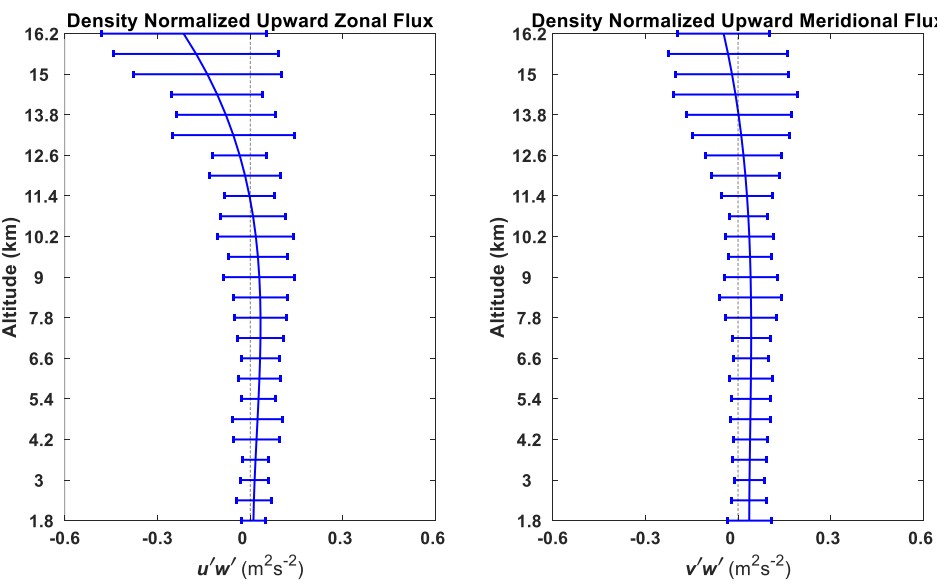

**Figure 16: Monthly averaged density normalized momentum fluxes**






We have estimated the horizontal momentum fluxes using the density derived from the radiosonde observations. Values lie in the range between ± 0.5 Pa. The largest values are generally associated with the bursts of wave activity evident in Figure 14. The mean monthly values are shown in Figure 17. Typical values are less than 0.05 Pa.

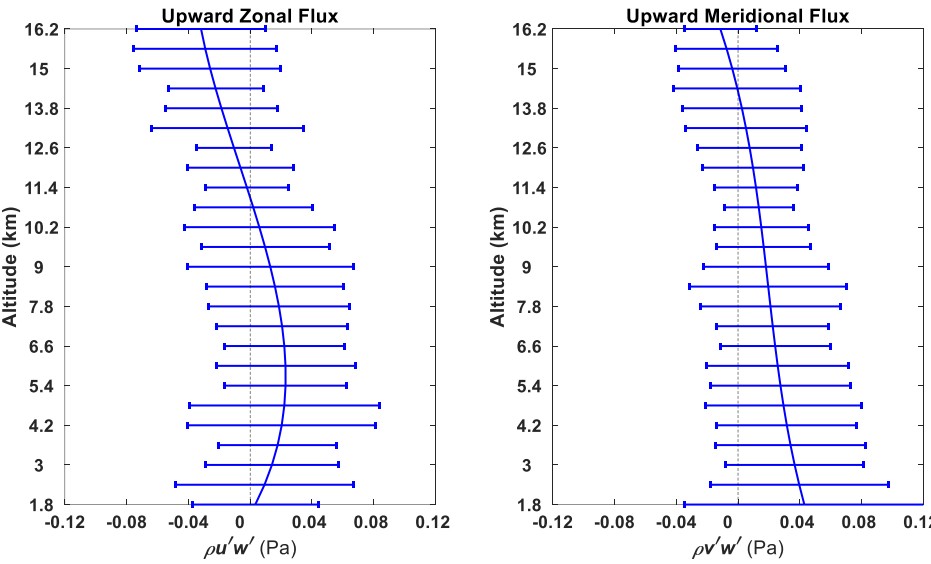


**Figure 17: Monthly averaged momentum fluxes**

### 4.2.2 Mean Flow Acceleration

The mean monthly mean flow acceleration is calculated with the equation (4) and is shown in Figure 18. $\bar{\rho}$, the mean neutral

atmospheric density has been obtained from the radiosonde measurements.

$$\widetilde{DF} = (DF_u, DF_v) = -\frac{1}{\bar{\rho}}\left(\frac{\partial \overline{\rho u'w'}}{\partial z}, \frac{\partial \overline{\rho v'w'}}{\partial z}\right) \tag{4}$$

Values are typically larger in the stratosphere than the troposphere. Mean values are generally less than about $3 \ ms^{-1}day^{-1}$.



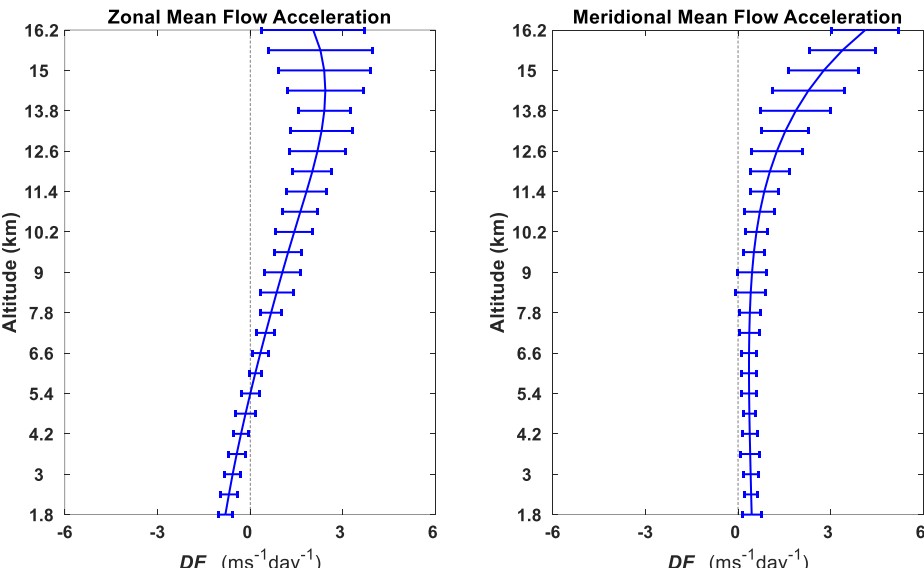

**Figure 18: Monthly averaged Mean flow acceleration**


## 4.3 Mesosphere Lower Thermosphere Winds

To calculate the meteor winds, 3-hourly averaged horizontal winds, stepped in hourly intervals, were determined from the measured radial velocities (Hocking et al., 2001), using uniform altitude bins of 2 km centered on 80 km to 98 km. Meteors that have zenith angles of less than 10° or more than 50° as well as those with a radial drift velocity greater than 200 m/s were

discarded. In order to remove outliers from the input radial velocity distribution, the iterative scheme proposed by Holdsworth et al. (2004) were applied, which involves performing an initial fit for the wind velocities, removing the radial velocities whose value differs from the horizontally projected radial wind by more than 25 m s$^{-1}$ and repeating the procedure until no outliers are found or until less than six meteors remain.

Two low-pass-filtered versions of the horizontal wind time series using an inverse wavelet transform with a Morlet wavelet

basis were calculated by applying the method proposed by Spargo et al. (2019). A 'narrow band' low-pass wavelet filter with a cut-off of 2 d has been applied to the hourly interval horizontal winds to evaluate the mean background winds. Since one-month long wind time series of each altitude was constructed for one time, a minimum scale size of 48 h and a total number of scales of 250 were selected. The resulting background horizontal winds are shown in Figure 19 as a height time plot and in Figure 20 as height profiles. The height time plots clearly include several periodicities, including a 3- to 4-day component.






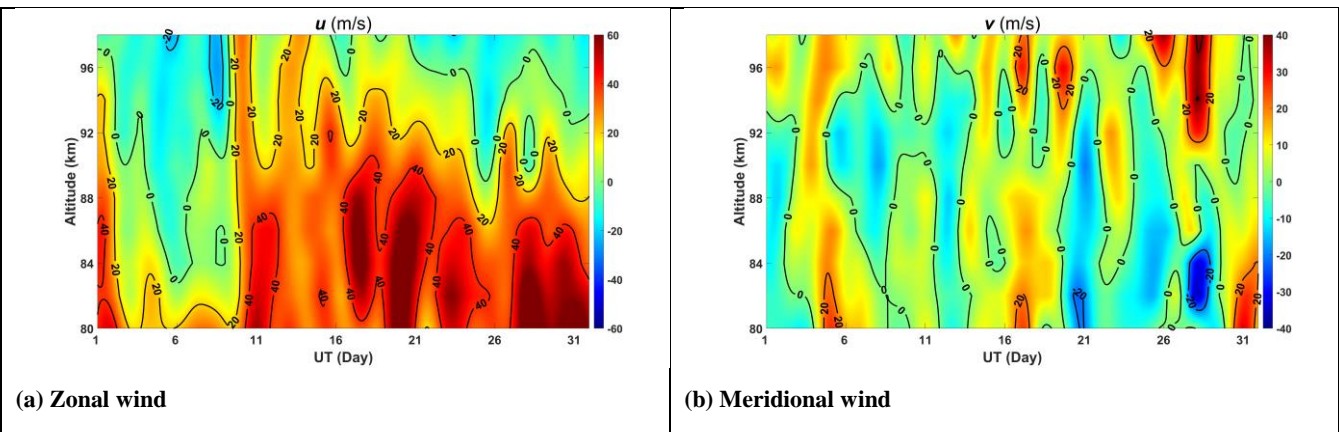

**(a) Zonal wind**

**(b) Meridional wind**

**Figure 19: The hourly interval background horizontal winds**

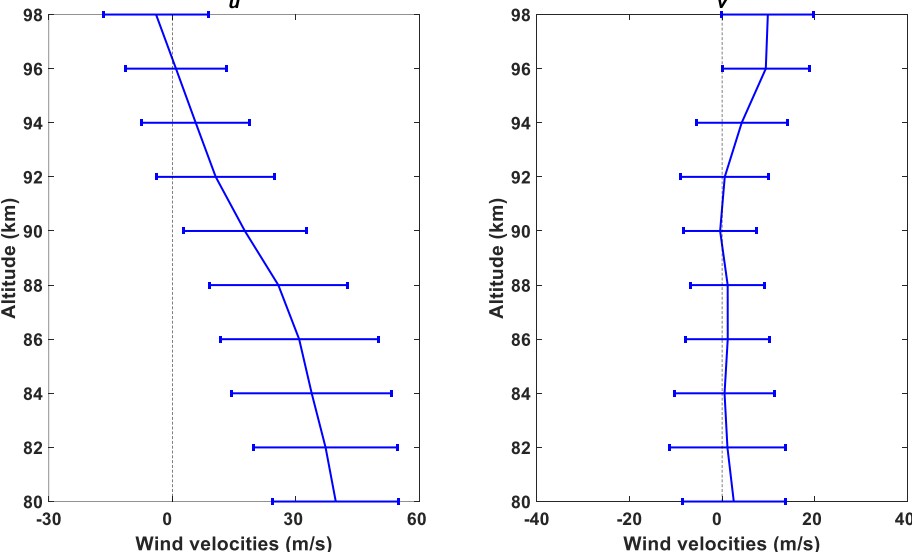

**Figure 20: The monthly mean horizontal winds**


### 4.3.1 Gravity Wave Momentum Flux and Vertical Transport in the Mesosphere Lower-Thermosphere Region

The density normalized GW momentum fluxes ($\overline{u'v'}$, $\overline{u'w'}$ and $\overline{v'w'}$) and kinetic energies ($\overline{u'^2}$, $\overline{v'^2}$ and $\overline{w'^2}$) in the MLT were derived following the method proposed by Hocking (2005) and subsequently improved by e.g. Spargo et al., (2019). To remove the influence of mean winds, long period planetary waves and tides, a 'broad band' low-pass-filtered version of the horizontal wind time series was calculated as well. A one month's wind time series of each altitude was reconstructed at one time, a





minimum scale size of 6 h and a total number of scales of 400 were selected to ensure that the filtered time series pertain to tidal-like (or longer) wind oscillations.

The reconstructed background winds were linearly interpolated between adjacent intervals to the time and height of each individual meteor echo within the given interval. The component of the value of the mean background wind along the meteor
line of sight was subtracted off the individual meteor's observed radial velocity to derive the residual velocity perturbation due to GWs. Covariances were then calculated from these residual perturbation velocities using the matrix-inversion method outlined in Hocking (2005). The radial velocity outlier rejection procedure following Spargo et al. (2019) was utilized to remove meteors with dubious square radial-velocity–AOA pairs from the input distribution to reduce the bias in the resulting covariance estimates. Unphysical results such as negative $\overline{u'^2}$, $\overline{v'^2}$, and momentum fluxes results ($\overline{u'w'}$ and $\overline{v'w'}$) with an
absolute value exceeding 300 m$^2$ s$^{-2}$ were discarded as well. Covariances were evaluated with 1-d window windows, with time shift of 6-h between adjacent windows. Monthly averaged winds and GW momentum fluxes were finally estimated.

### 4.3.2 Density Normalized Reynolds Stress Terms

Height time plots of the density normalized Reynolds stress terms are shown in Figure 21. There is considerable structure evident in these plots. However, generally, they show a similar modulation of about 4-days to that evident in the background
winds shown in Figure 19. This is particularly so for the $\overline{u'w'}$ and $\overline{v'w'}$ terms. The vertical kinetic energy and the upward fluxes of horizontal momentum fall off in value until about 86 km, where they then change little in value with height.

### 4.3.3 Monthly Averaged Reynolds Stress Terms

The monthly-averaged Reynolds Stress terms are shown Figure 22. $\bar{\rho}$, the mean neutral atmospheric density has been obtained from the MSIS00 atmospheric model. The horizontal kinetic energy terms decrease with increasing height, indicating the loss
of wave energy with height.

We can see that fluxes predominantly decrease with increasing height, and monthly mean zonal winds decrease with increasing height and reverse above 96 km. The monthly-averaged meridional momentum fluxes are predominantly southward, opposite in sign to the monthly mean meridional winds which are dominated by the week northward flow.

### 4.3.4 Mean Flow Acceleration

Mean flow accelerations were calculated with equation (4) and are shown as height profile time plots in Figure 23.





**Figure 21: the density normalized Reynolds Stress terms: (a) $\overline{u'^2}$ (top left), (b) $\overline{v'^2}$, (top right), (c) $\overline{w'^2}$ (centre left), (d) $\overline{u'v'}$ (centre right), (e) $\overline{u'w'}$ (bottom left), and (f) $\overline{v'w'}$ (bottom right).**

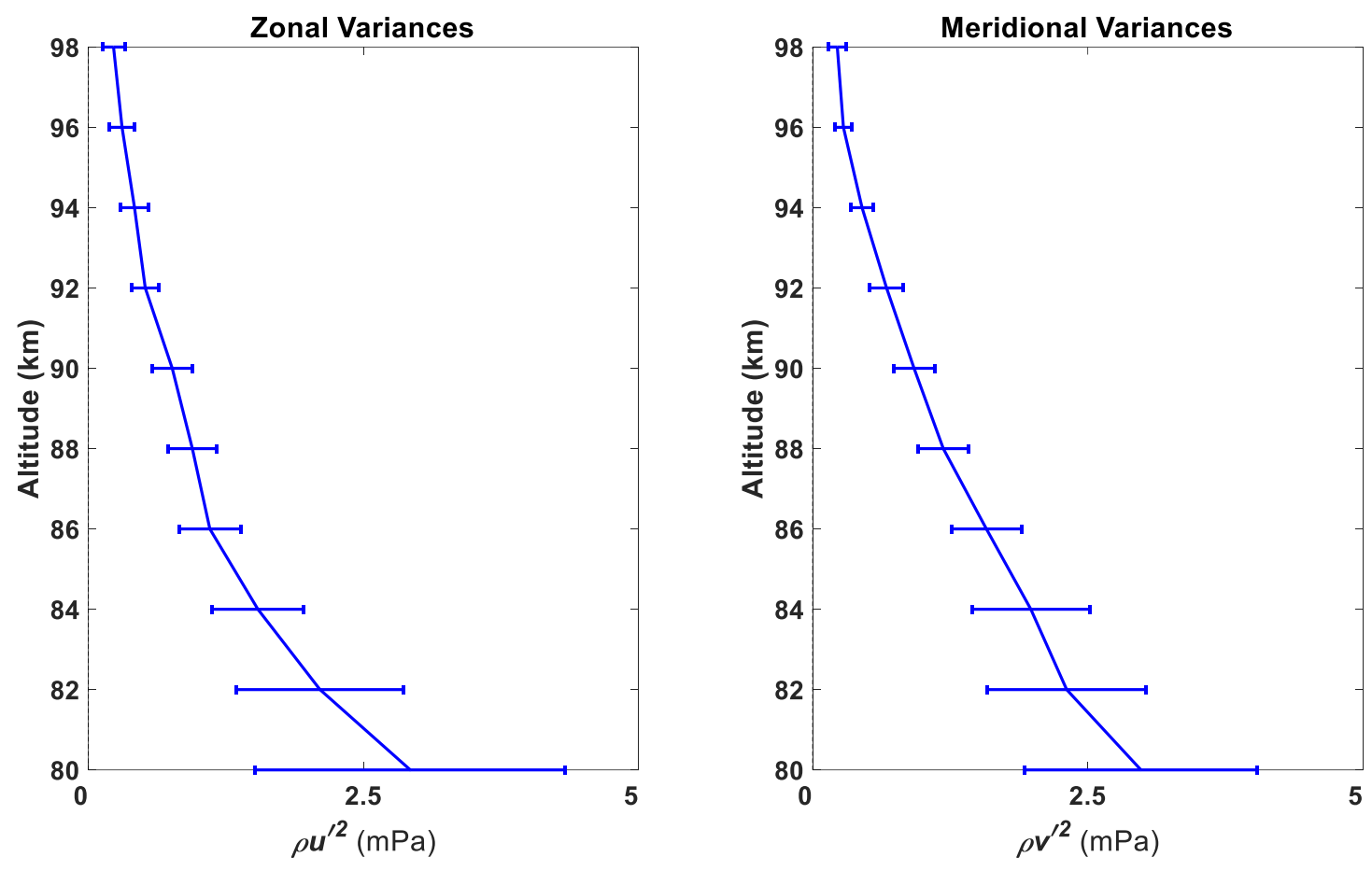

(a) Monthly averaged $\overline{\rho u'^2}$ and $\overline{\rho v'^2}$

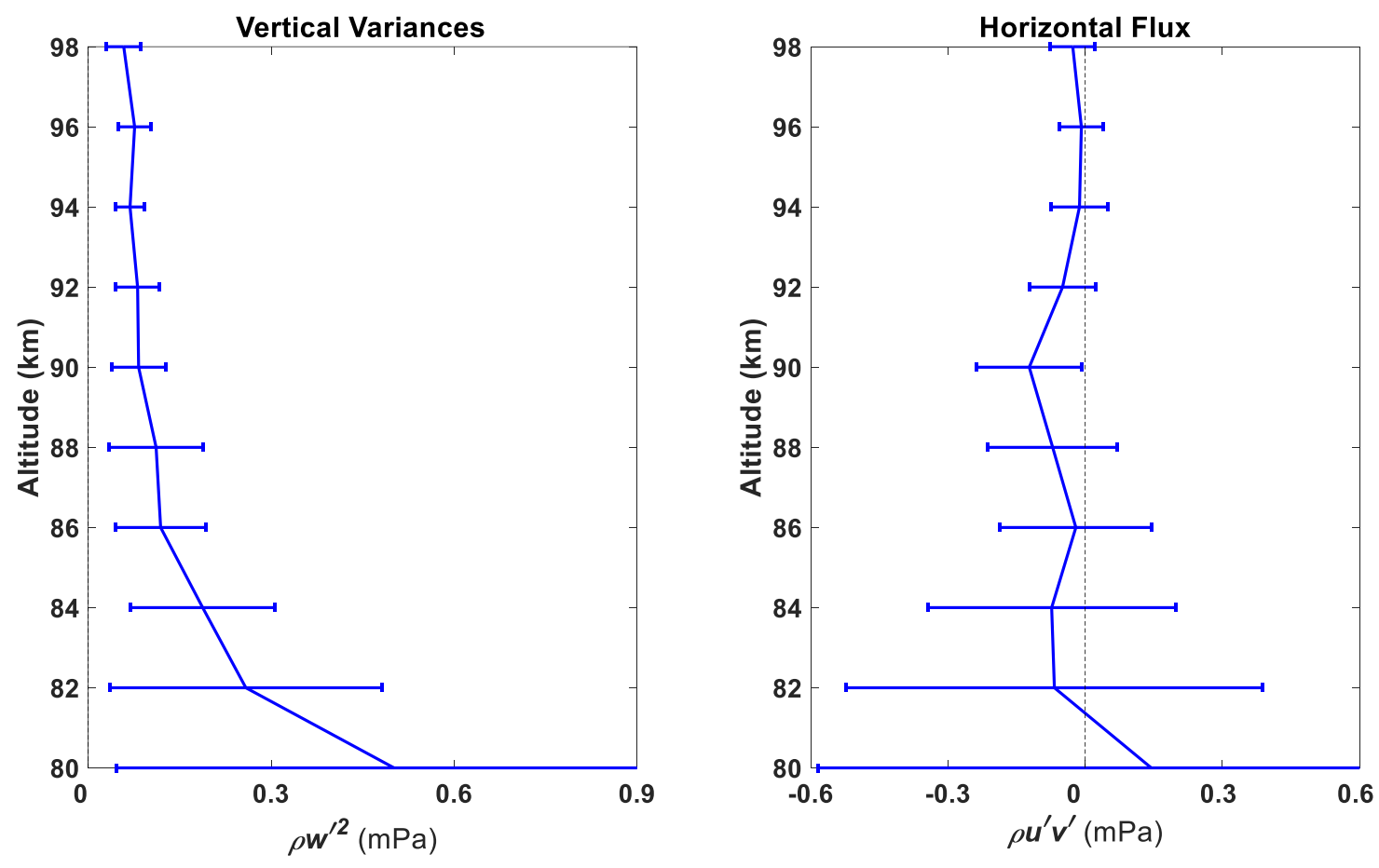

(b) Monthly averaged $\overline{\rho w'^2}$ and $\overline{\rho u'v'}$





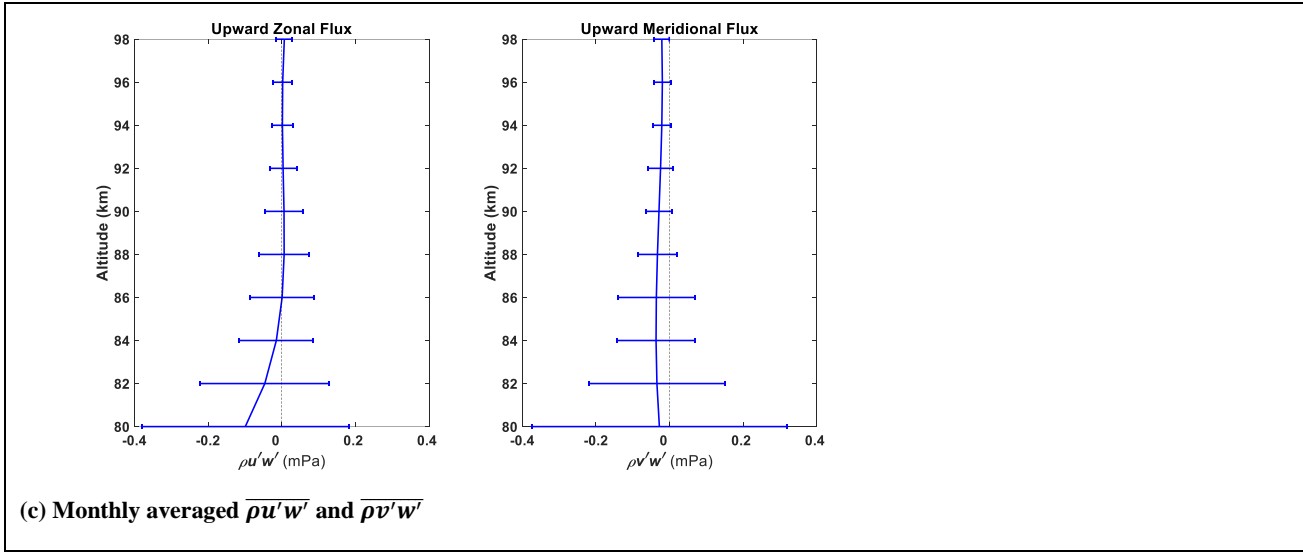

**(c) Monthly averaged $\overline{\rho u'w'}$ and $\overline{\rho v'w'}$**

**Figure 22: Monthly mean profiles of the Reynolds Stress terms**

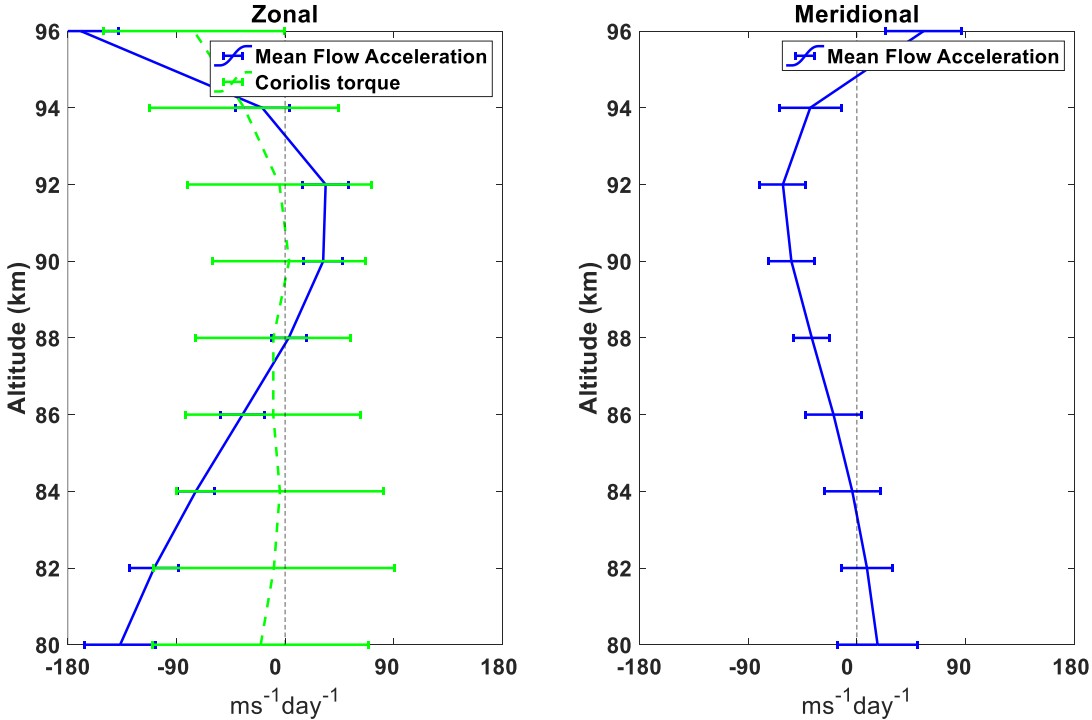

**Figure 23: Monthly averaged Mean flow acceleration**





The components of the monthly averaged mean flow acceleration are shown in Figure 23 along with the Coriolis torque due to the local meridional wind. This has been reversed in sign for ease of comparison. Values of the mean flow acceleration are easily large enough to balance the Coriolis torque in the zonal case, and have a similar form.

## 4.4 Integrated Wind Observations

The new radar provides winds in the troposphere and stratosphere and in the mesosphere and lower thermosphere and it is
interesting to consider the winds available above the radar. Figure 24 and Figure 25 show the 6-h interval horizontal background winds of the Dual-Frequency ST / Meteor Radar covering heights from near 1 km to 16 km and from 80 to near 98 km, along with 12-h interval horizontal background winds of radiosonde covering heights from 16 km to 30 km. These demonstrate the true 'MST' capability of the new radar. With the addition other equipment, such as a Rayleigh Doppler Lidar also built by NSSC (see e.g., Yan et al., 2017), there is potential to fill the radar 'gap' region and provide a continuous wind
profile from the ground to heights near 100 km fulfilling one of the goals of the original vision for MST radars.

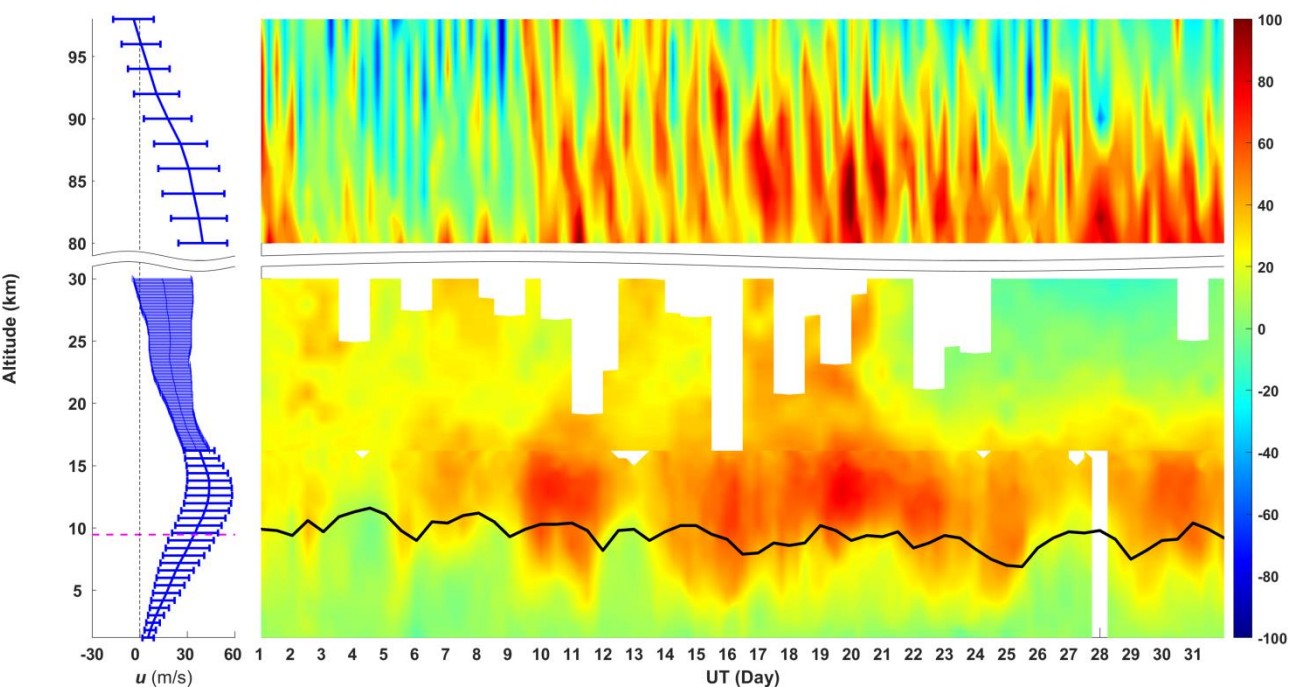

**Figure 24: zonal wind, pink line(left) and black line(right) indicate tropopause height.**





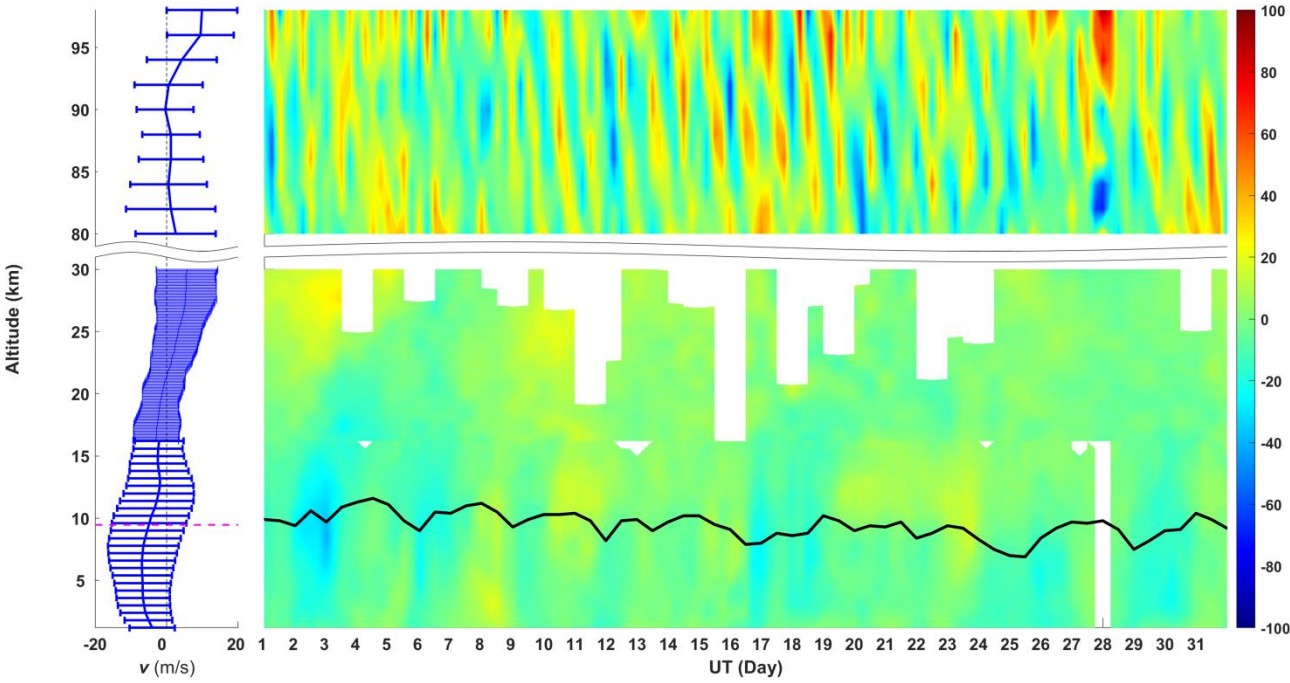


**Figure 25: meridional wind, pink line(left) and black line(right) indicate tropopause height.**

## 5 Summary and Outlook

The present paper gives the description of the novel system design and functionality of the basic components of the new dual-

frequency ST / Meteor Radar installed in Langfang Observatory of NSSC. Two frequencies (53.8 MHz and 35.0 MHz) are

used in interleaved operation thus optimizing performance for both ST wind retrieval and meteor trail detection. The new radar

exhibits true MST capability from its first results. In solo meteor mode, the daily meteor count rate reaches over 40,000 and

hence readily allow wind estimation at the finer time resolution of 30-min better than the 1-hour typical of most meteor radars,

demonstrating the potential to investigate shorter period motions in the mesosphere. The uncertainty of the ST wind

measurements is better than 2 m/s when estimating the line of best fit with radiosonde winds launching about 50 km north to

the radar site. We also present the preliminary observation results of typical winter GW momentum fluxes in the TLS and

MLT of Langfang district. Intense GW activities were found during a cold front beginning on Dec. 16th and appear trapped

near the tropopause. On the other hand, modulation between GWs and planetary waves are evident in the MLT. Campaigns

with other equipment, such as Rayleigh Doppler Lidar, may give us more integrated wind information from the ground to

heights near 100 km, reveal more wave activities, and contribute to the study of atmospheric dynamics.



*Code / data availability*. The radar data in this study is available on request from Qingchen Xu.

*Author contributions*. Qingchen Xu was in charge of the installation and operation of the radar, and carried out the data analysis. I. M. Reid contributed to the data analysis. Bing Cai contributed to the data analysis and diagram drawing. Christian Adami supervised the installation and test running of the radar. Zengmao Zhang, Mingliang Zhao and Wen Li contributed to the installation and test running of the radar. Qingchen Xu and I. M. Reid wrote the paper.

*Competing interests*. The new radar introduced in this study was designed and manufactured by ATRAD Pty. Ltd. Iain Reid is the executive director of this company, and C. Adami is the engineer of this company.

*Acknowledgements*. This work was supported by the Strategic Priority Research Program of Chinese Academy of Sciences (Grant No. XDA17010302). The radiosonde data used in this paper were provided by Beijing Meteorological Observatory.

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
