# Peer review of "A New Dual-Frequency Stratospheric Tropospheric / Meteor Radar: System Description and First Results"

_Atmospheric Measurement Techniques, 2023_

## Author Comment (AC2)

Reply to RC2

**Thanks for the very detailed comments and suggestions. Now we will reply to these comments one by one.**

*General comment:*

*The manuscript presents the initial results of a recently installed ST/meteor radar at Langfang, China. The radar uses a dual-frequency transmitter and receiver system layout for interleaved operation in meteor and stratosphere-troposphere measurement mode. The system performance is demonstrated by wind observations. The stratospheric-tropospheric measurements are cross-compared to radiosondes. There are also results shown deriving gravity wave momentum fluxes. However, there are some concerns related to the analysis and conclusions of how this analysis. The manuscript fits well into the scope of the journal and will become publishable after addressing some concerns.*

**Thanks for the comments.**

*Figures 16 and 17:*

*These Figures are a duplication of content and should be merged by including a second x-axis on top and a different line color.*

**We have merged these figures following the reviewer's suggestion.**

*Line 306-308:*

*The sentence seems to refer to Figure 15. However, this Figure shows the momentum fluxes as  in m2/s2. Please change the sentence that is matches the units shown in Figure 15?*

**We have revised this part following the reviewer's suggestion.**

*Section 4.2.2.*

*This section contains three lines and one equation. Figure 18, which the reviewer assumes, belongs to this section is not mentioned. The reviewer suggests*

*removing this section entirely or expanding this part. This section appears to be disconnected from the main narrative of the paper. In its present form it cannot be published. However, the topic is scientifically important and might can be investigated in the future.*

**Since the mean flow accelerations of TLS and MLT region in this manuscript are calculated with the equation (4), we have supplemented the content of section 4.2.2 and merged sections 4.2.1 and 4.2.2 into one following the reviewer's suggestion. More study will be fulfilled on this topic in the future.**

*Lines 339-345:*

*This paragraph is very confusing. At first, different wavelet filters are described, but later Figure 19 and 20 show monthly mean and hourly mean winds? It is recommended to add information when the measurements were taken. The use of Figure 20 is unclear in the context of this paragraph. How do the mean winds compare to the other observations nearby e.g, Mohe, Wuhan?*

*The reviewer assumes that this section is supposed to path the way for the momentum flux analysis, it is recommended to show the original hourly time series with all waves included and the two filtered data products.*

**Following the reviewer's suggestion, we have supplemented 'raw' and 'low-pass-filtered' background horizontal winds in original Fig. 19 to path the way for the momentum flux analysis. The content of original Fig. 20 now is combined in original Fig. 22 together with monthly averaged Reynolds stress terms and mean flow acceleration.**

*Sections 4.3.1 to 4.3.4:*

*GW momentum fluxes inferred from meteor radar observations are not straightforward. The most critical aspect is the vertical wind, which is severely biased. A simple least square fit might be not sufficient and more sophisticated mathematical approaches seem to be necessary (https://amt.copernicus.org/articles/15/5769/2022/amt-15-5769-2022.html). As described in the paper, some least square fits result in negative values of the Reynolds-stress tensor on the main diagonal. The momentum fluxes shown in Figure 21 reach values of up to 150 m/s. When applying equation (4), this would lead to mean flow accelerations in the order of 1000 m/s/day for the instantaneous observation. The reviewer suggests discussing these results with*

*other measurements in the literature (e.g., https://doi.org/10.5194/angeo-33-1091-2015,https://doi.org/10.1002/2014GL060501, https://doi.org/10.1002/2016GL068599, https://doi.org/10.1002/2016GL072311). It would be also good to calculate the cooling rate that is associated with such a mean flow acceleration. The reviewer also suggests merging all four sections into one. Figure 22 is not needed.*

**We agree that GW momentum fluxes inferred from meteor radar observations are not straightforward as the reviewer has mentioned, and we should consider taking the methods proposed by e.g. Stober (2022) in the following study as the reviewer has pointed out.**

**In this manuscript we followed the method proposed by Hocking (2005) which is the same in the papers refereed by the reviewer. However, those authors have applied much longer integration time for the estimation of GW momentum fluxes, such as 10-day moving average or monthly average. We applied 1-day average in this manuscript. Large values such as near 150 m/s (which is apparently noise owing to the relatively short integration time as Spargo has pointed out in Spargo 2019) are shown in original Fig.21.**

**Following the reviewer's suggestion, GW momentum fluxes are estimated again with 10-d moving average. They are much smoother than previous results and values of momentum fluxes are less than 60 $m^2/s^2$ in most cases, which are typical and similar with the results in the papers refereed by the reviewer. However, the modulations between GWs and PWs are not as evident as in the previous results. We comment that shorter averaging periods than the 10-d period used here show very suggestive structures in the upward flux terms that may be related to some of the planetary wave activity evident, but they are also somewhat noisy and show magnitudes that are likely much too large. We will investigate this in future work.**

**Original Fig.22 now is combined with monthly averaged horizontal winds and Mean flow acceleration and Original Fig.23 is removed. Section 4.3.1 to 4.3.4 are now merged into one.**

*Line 399-400:*

*Why not fill the gap region with radiometric temperature and wind observations, which are less sensitive to tropospheric clouds?*

**Original Figs.21 and 22 were to demonstrate the true 'MST' capability of the new radar. Winds of radiosonde were used to evaluate the ST wind measurements, and together we put them in the same plots to show the**

**data we have used in this paper. There are many ways to fill the gap region, such as radiometric temperature and wind observations, and Rayleigh Doppler Lidar that we mentioned in the last section. Multiple-instrument campaigns are anticipated and we'd like to collaborate with other groups in the future work.**

*Conclusions/summary/outlook:*

*The conclusions should be revised. The new system is valuable, and the paper demonstrates very good science opportunities. These achievements should be emphasized in this section.*

**We have revised this part following the reviewer's suggestion.**

**Minor comment:**

*Line 12: 'novel' – new is the better term as it is used in the title. Dual frequency operation with shared hardware is not entirely 'novel'.*

**We have revised this part.**

*Line 37:  'the' is too much …regions with one radar.*

**We have revised this part.**

*Line 45: remove 'directly by'*

**We have revised this part.**

*Line 46: cite Hocking, 2005 here*

**We have revised this part.**

Figure 2: The figure Quality is low

*Line 107: The ST-radar antenna array consists of linear polarized antennas?*

**Yes, and we have supplemented descriptions.**

*Line 119: PRF of 200 kHz (700m monostatic range)???*

**The pulse repetition frequency (PRF) can be set up to 200 kHz for this radar. We now set the PRF to 14 kHz for ST Low Mode (see Table 2) and lower PRF for other modes.**

*Line 135: accumulated – may be collected.*

**We have revised this part.**

*Line 148-150: The spatial averaging of about 400 km in diameter removes many of the small-scale waves and thus limits the benefit of a 10- or 15-minute temporal resolution.*

**Certainly, you are right. However, Figs. 5 and 6 do exhibit good continuity with 30-min intervals and show more detailed variations than the 1-hour interval data.**

*Figure 8 and 9 should be reduced. It is maybe sufficient to show one Figure with a few days (or a month) comparing both wind components.*

**Figs. 8 and 9 are reduced now with wind comparisons over one month.**

*Line 178-180: maybe reference:*
*https://angeo.copernicus.org/articles/35/893/2017/*

**We have added the reference.**

*Line 223: Radar profiles from 30 minutes before …..*

**We have revised this part.**

*Line 264:265: Please add whether these velocities are the line-of-sight measurements of the vertical beam or fitted data from all five beams. Please also include in this section how the spectra are analyzed (moments method vs. fit?) and whether additional coherent or incoherent integrations are added. Also information of the dwell time can be summarized here.*

**We have revised this part.**

*Line 358-361:   Maybe reference: https://angeo.copernicus.org/articles/39/1/2021/angeo-39-1-2021.html*

**We have added the reference.**

*Line 365: one 'window' too much*

**We have revised this part.**